# Observations for Sjögren’s Pigment Epithelial Reticular Dystrophy in a 16-Year-Old Boy—An Extremely Rare Retinal Case Report

**DOI:** 10.3390/jcm12041406

**Published:** 2023-02-10

**Authors:** Monika Modrzejewska, Wojciech Lubiński, Katarzyna Czyżewska, Wiktoria Bosy-Gąsior

**Affiliations:** 1II-nd Department of Ophthalmology, Pomeranian Medical University in Szczecin, Al. Powstancow Wielkopolskich 72, 70-111 Szczecin, Poland; 2Departament of Neonatal Intensive Care Unit, Pomeranian Medical University in Szczecin, Al. Powstancow Wielkopolskich 72, 70-111 Szczecin, Poland; 3Scientific Association of Students II-nd Department of Ophthalmology, Pomeranian Medical University in Szczecin, Al. Powstancow Wielkopolskich 72, 70-111 Szczecin, Poland

**Keywords:** Sjögren’s reticular dystrophy, degenerative lesions in the retinal pigment epithelium, diagnostics, electrophysiological changes, ophthalmic symptoms

## Abstract

The purpose of this publication is to present an extremely rare case of Sjögren’s pigment epithelial reticular dystrophy. So far, 10 such publications have been found in world literature. A 16-year-old boy was diagnosed due to a slight loss of visual acuity, confirmed in static perimetry/24-2/. Abnormal dense clusters of retinal pigment epithelium (RPE) cells forming a reticular network pattern (resembling a fishing net) with marked knots were detected by fundoscopy in the macular area and the mid-periphery of the retina. No abnormalities were found in the anterior segment, intraocular pressure, kinetic perimetry, Ishihara or Farnsworth D-15 tests or OCT. Fluorescein angiography confirmed blocked fluorescence from the choroidal vessels caused by the pigment in RPE. An autofluorescence test showed hypofluorescent foci corresponding to symmetrical and bilateral retinal hyperpigmentation with an RPE reticular pattern. Multifocal ERG (mfERG) revealed slight cone photoreceptor and bipolar bioelectrical dysfunction. Electrooculography (EOG) showed significant asymmetry (Arden Ratio 1.8), suggesting bioelectrical dysfunction of RPE/photoreceptors. Flash ERG (ERG) revealed only slight increase in implicit time of the a and b waves of the rod and cone responses and exclude cone-rod dystrophies. This article highlights the importance of the results of ophthalmoscopy, fluorescein angiography, autofluorescence, mfERG, fERG, EOG and genetic tests for Sjögren’s reticular dystrophy with a pathogenic variant in the region of the C2 gene-c.841_849+19del (dbSNP rs9332736).

## 1. Introduction

Sjögren’s reticular dystrophy is an extremely rare disease. It is characterized by a network of hyperpigmentation resembling a fishing net with knots—first involving only the posterior pole, then moving to the periphery with a presumably unaffected choriocapillaris [1]. This disease belongs to the group of pattern dystrophies of the retinal pigment epithelium (RPE), which also includes, inter alia, butterfly-type pattern dystrophy, adult-onset foveomacular vitelliform dystrophy and fundus pulverulentus. Studies show that the clinical picture (e.g., fundus examination) in this group of diseases may differ among affected family members or between two eyes of the same person.

Over time, one pattern may also evolve into another [2]. Pattern dystrophies are assigned mutations in the human retinal degeneration slow (RDS) and peripherin gene on chromosome 6 at position 21.1. The prognosis for this group of diseases is usually optimistic, although sometimes they may be accompanied by a gradual loss of vision [3].

In this report, we describe the clinical history of a 16-year-old boy diagnosed with reticular dystrophy of the RPE. Not many case reports have been found in the global literature, making the case we describe unique. So far, in 70 years of observation, only 10 publications on confirmed Sjögren’s pigment epithelial reticular dystrophy (and 3 articles on suspected cases) have been found in world literature (PubMed database). The first such was described in 1950 by Henric Sjögren [4].

## 2. Case Description

A 16-year-old healthy boy visited an ophthalmologist due to a 6-month period of distance vision blurring. An ophthalmological examination at the local outpatient clinic revealed degenerative lesions in the macular area of the retina, and therefore Stargardt’s disease was suspected. The patient was referred to the Department of Ophthalmology for more detailed diagnostic procedures. Examination for distance and near visual acuity showed normal parameters for the best corrected visual acuity (Snellen charts): VRE (right eye) 0.9 sc;cc-0.25D cyl ax 110 = 1.0 and VLE (left eye) 0.9 sc;cc-0.75D cyl ax 2 = 1.0. In addition, for near vision: VRE 1.0 cc+0.5D/−0.25D cyl ax 110 and VLE 1.0 cc+0.5D/−0.75D cyl ax 2, but static perimetry examination confirmed a subtle disturbance of sensation in the central retina/24-2/in both eyes, values within limits (RE: MD −2.36 dBl *p* < 5%; PSD 1.62 dB and LE: MD −2.55dB *p* < 5%; PSD 1.51) [Figure 1].

No abnormalities were found during the examination of the anterior segment of the eye, intraocular pressure, Ishihara test for color vision or Farnsworth D-15 color test. Intraocular pressure in both eyes was normal: 16.5 mmHg for the RE and 17.2 mmHg for the LE (ICARE tonometer). In the eye fundus, indirect ophthalmoscopy by a Volk lens revealed symmetrical bilateral lesions with a characteristic pattern in the form of a circular rearrangement of RPE cells around the optic disc (foci size approx. one disc diameter) and hypertrophic hyperpigmentation around the macula extending to the periphery of the retina, forming a characteristic reticular pattern (resembling a fishing net) towards the perimeter (zones 2, 3). The retinal vessels were normal. There were no structural anomalies of the retina in macular OCT: central retinal thickness RE 282 μm and LE 279 μm. No anomalies in kinetic perimetry and no deficiencies in color perception were noted in color vision tests (Ishihara test and Farnsworth D-15 test) [Figure 2].

The degenerative lesions in the retina in the form of hyperpigmentation of RPE cells and their characteristic image and location at the fundus of both eyes were confirmed by autofluorescence imaging as numerous patchy or mosaic foci of hypofluorescence [Figure 3 and Figure 4].

Reticular pattern of pigment epithelial hyperpigmentation morphologically like a fishing net with knots.

Fluorescein angiography detected blocked fluorescence from the choroidal vessels caused by the present pigmented lesions, which is confirmed and clearly visualized by the characteristic reticular pattern [Figure 5].

Multifocal electroretinography (mfERG) was normal in the left eye but abnormal in the right eye. In addition, the increase of implicit time of P1—wave from rings R1–R3 indicates slight macular cone photoreceptors/bipolars dysfunction secondary to RPE degeneration in this region [Figure 6]. 

The EOG test result was normal in both eyes, however, significant asymmetry was shown (Arden ratio: RE 3.3 and LE-1.8). Near the lower limit of the normal Arden ratio in the left eye suggested possible bioelectrical dysfunction on the level of external layers of the retina (RPE/photoreceptors) [Figure 7].

Pattern ERG was normal in both eyes. This comes as no surprise as this is a response mainly from ganglion cells which are not involved in reticular dystrophy during the early stages of the disease [Figure 8].

It is sensible to perform flash ERG (ERG) in suspected reticular dystrophy because this test can help in the differential diagnosis of cone and cone-rod dystrophy. The only slight increases of implicit times of the a and b waves of the rod-cone response, exclude in high probability cone and cone-rod dystrophies.

In addition, for differential diagnosis, genetic tests of the NGS (Next Generation Sequencing) type were performed. The sequence of 317 gene coding regions was analyzed, and the pathogenic variants correlated with the occurrence of retinitis pigmentosa or retinal diseases according to data contained in the RetNet (Retinal Information Network) database and the literature data [5]. The patient has not been diagnosed with mutations in the above genes. The result, however, confirmed the presence of the pathogenic variant in the area of the C2 gene-c.841_849+19del, dbSNP rs9332736. This lesion has not been described in the literature related to retinal dystrophy diseases. Therefore, the test result does not confirm but also does not exclude the clinical diagnosis of any retinal diseases covered by the study.

The patient’s mother and sister had been ophthalmologically assessed in the past, but ophthalmoscopic examinations did not reveal degenerative lesions of the retina in the fundus of the eye. However, additional ophthalmological examinations were not performed on them to confirm pigment epithelial reticular dystrophy, which may pose a limitation in this case report.

However, we know that with Sjogren’s syndrome, the image of the fundus among family members may vary or even be slightly marked. As a result, extensive retinal examinations are recommended.

## 3. Discussion

Reticular dystrophy of the RPE was first described by Sjögren in 1950 in eight of thirteen siblings from a Swedish family. The hallmark of this condition is the presence of a symmetrical, bilateral reticular network with pigment cells arranged in clusters in the reticular pigment epithelium. These anomalies are accompanied by degenerative lesions of the pigment epithelium (hypertrophy and dystrophy) that often form a pattern resembling a fishing net, which changes the RPE function in EOG tests. The pattern of lesions resembled a fishing net with knots, similar to that in our patient [4,6]. Changes in the EOG, according to some authors, are related to the mode of inheritance [7].

According to Deutman et al., after spreading towards the periphery of the retina, hyperpigmentation tended to disappear in previously observed retinal locations and reappear in others not yet occupied in later years [8,9]. The other authors presenting cases of Sjogren’s dystrophy present architectural diseases in the natural course rearrangement of the dye in the RPE by a kinetic process in successive time steps, but little is known about the late stages of this disease [10].

Deutman and Rumke’s 14-year-old patient had a normal EEA result. Other authors have also reported normal or borderline EEA. Our patient presented mild changes in the EOG in one eye, suggesting an early stage of reticular dystrophy in the RPE, with good visual acuity.

Normal visual acuity is emphasized by many authors; this parameter is likely related to the autosomal dominant mode of inheritance [8,10].

Although pattern dystrophies rarely damage the function of the RPE and do not impair vision, isolated cases of this disease have been reported due to subretinal neovascular membranes, as described by Shiono et al. [7].

Sjögren’s reticular dystrophy can be inherited in an autosomal dominant or recessive pattern. It is characterized by a symmetrical, bilateral reticular network with pigment cells arranged in clusters in the reticular pigment epithelium. These anomalies are accompanied by degenerative lesions of the pigment epithelium (hypertrophy and dystrophy) that often form a pattern resembling a fishing net [4,8]. Visual acuity remains unaffected or only mildly abnormal in advanced stages, although an association of Sjögren’s reticular dystrophy with choroidal neovascularisation has been reported. Peripheral fields and color vision can be normal or blurred, which may be confirmed as the absence of functional anomalies of the retina in ERG and EOG tests [4], contrary to our observations, where the EOG and mfERG tests revealed slight but noticeable changes in the initial functions of cones and RPE, as stated by other authors [4,6,7,10]. Typically, the EOG test is often somewhat abnormal in patients with reticular dystrophy, but not in all cases. Other authors have reported normal, borderline, or subnormal EOGs [10].

In the presented case, the EOG was not helpful in confirming the diagnosis, although the lower Arden value with the asymmetry was found in the left eye, which could be taking into account in initial differentiation in the level RPE/photoreceptors, given the young age of the patient and the early stage of the disease. ERG test results indicate that the neurosensory retina is either minimally or not involved in the disease process. The described changes are correct or almost correct; only Fishman et al. describe two cases—photopic and scotopic changes in the ERG [11].

In the case we analyzed, it should be emphasized that there were slight disturbances in the ERG tests, which may be related to the pathogenic variant in the region of the C2 gene-c.841_849+19del (dbSNP rs9332736) confirmed in the patient, which has not been mentioned in the literature so far. Multifocal ERG (mfERG) revealed discreet dysfunction of the macular cones photoreceptors and bipolars. With electrooculography (EOG), significant asymmetry was shown (LE- Arden Ratio 1.8 ), suggesting slight bioelectrical dysfunction of RPE/photoreceptors. Flash ERG (ERG) revealed only slight increases in implicit times of the a and b waves of the rod-cone response, and exclude in high probability cone and cone-rod dystrophies.

Fluorescein angiography plays an important role in diagnosis, enhancing the characteristic mosaic pattern with hyperpigmentation in the RPE and indicating abnormal pigmentation and blocked fluorescein migration from the choroidal vessels associated with the presence of lipofuscin within the reticular pattern, despite the normal structure of capillary choroidal vessels of larger diameters, similar to the case report [6,8,10]. In later years, Deutman et al. reported their observations on retinal lesions in four brothers who had butterfly-shaped pigment dystrophy in the macular region. They emphasized that the hyperpigmentation areas extended to the sensory retina, including the retinal pigment epithelium [9], which is in line with our observation.

In the initial stages, pigment grains accumulate in the fovea, and with time they form a network around central lesions, extend towards the foveal periphery and create a characteristic pattern. The areas between network lines arranged around the dark pigmented area in the macula are irregular, similar to the characteristic image of autofluorescence imaging of the presented patient. This network is mildly to moderately hyperautofluorescent on autofluorescence imaging and bright on near-infrared reflectance imaging. Optical coherence tomography showed abnormalities of the retinal pigment epithelium—Bruch membrane complex, photoreceptor outer segments, and photoreceptor inner/outer segment interface.

Their size does not exceed one optic disc diameter, and the reticular network extends in all directions in areas 4–5 times the optic disc diameter from the macula. The central and peripheral retina might be unaffected, but in some cases, they are the main locations of lesions [9]. According to Schauwvlieghe et al., reticular dystrophy is the result of the accumulation of both pigment and lipofuscin between photoreceptors and retinal pigment epithelium, as well as within the retinal pigment epithelium [12]. The retinal vasculature does not show tightness and indicates the outflow of the liquid from the vessel wall. The appearance of the optic disc and other areas of the fundus were normal, as were the results of retinal function tests.

Moreover, our patient developed a discrete blurring of vision in OP and disturbances both in mfERG and EOG, in RE and LE, respectively, with cone photoreceptor and bipolar bioelectric dysfunction and complex pigment epithelium (RPE) in the retinal outer receptor layer. So far, there has been limited data on changes in electrophysiology concerning this dystrophy in the literature. The described lesions are visualized during fluorescein angiography due to the contrast between the hyperpigmented and dystrophic areas. The reticular network pattern probably begins to form in infancy and is fully developed by 15 years of age. Pigmentation may disappear in older patients. When establishing the diagnosis, attention is drawn to the characteristic features of RPE cell hypertrophy, the normal parameters of visual acuity and the functionality of retinal photoreceptors.

In the case we analyzed, the mutations described in retinal dystrophies were not found [5]. So far, while the correlation of the disease with autosomal recessive and dominant inheritance has been observed, the available literature does not identify a clear gene responsible for the development of Sjogren’s retinal dystrophy [5,12]. In the description of two cases, Schauwvlieghe et al. also did not show the relationship between retinal dystrophy and the most frequently observed mutations in the PRPH2 and ABCA4 genes, similar to our patient [12]. It is possible that the presence of a pathogenic variant in the region of the C2 gene-c.841_849+19del (dbSNP rs9332736)—may be important in the development of this disease, which has not been discussed in the literature so far. Citations indicate a strong relationship between the gene mutation detected in our study and the development of autoimmune diseases, such as SLE or Sjogren’s syndrome [13]. Furthermore, there are recent reports of a correlation with the development of AMD, which has not yet been confirmed [14]. Therefore, we believe that the case we describe, which manifests typical symptoms of Sjogren’s retinal disease, may be an example of a different, more complex inheritance than previously speculated, the more so that it meets the recognition criteria given in the Rinaldi et al. article [1].

Differential diagnosis should take into consideration other hereditary retinal and choroidal dystrophies in which pigment lesions might be detected in the ocular fundus. These should include Type 1 myotonic dystrophy, Stargardt’s disease, Malattia Leventinese (Doyne honeycomb retinal dystrophy), Type II membranous-proliferative glomerulonephritis, North Carolina macular dystrophy, Benign concentric annular macular dystrophy (BCAMD), Retinitis pigmentosa, generalized cone-rod dystrophies. In order to rule out the cone and cone-rod dystrophy and to facilitate differentiation into reticular dystrophy, the fERG is an appropriate test that should be performed. Therefore, fERG was performed as well, and the results obtained by the authors exclude cone and cone-rod dystrophies with high probability.

So far, the treatment of Sjogren’s retinal dystrophy has not been clearly described in the literature, and symptomatic treatment is individually tailored to the stage and clinical picture of the disease. Recent research by Nguyen et al. reports on the possibility of using modern nanomedicine technology, which may, in the future, enable the treatment of hard-to-reach regions of the eyeball, such as the retina and its dystrophies carboxyl-terminated transacting activator of transcription cell-penetrating peptide (T) and metformin (M). In vitro, nanotherapeutics have been shown to possess persistent drug release profiles, good ocular biocompatibility, and potent bioactive activities for targeting prevailing risk factors associated with retinal diseases. Further advantages of these compounds lie in preventing the loss of antioxidants as well as inhibiting the growth of abnormal vessels [15]. This discovery may turn out to be revolutionary both in diseases for which treatment is currently known as well as in rare diseases for which therapies are still insufficient or lacking, such as Sjogren’s retinal dystrophy.

## 4. Conclusions

Sjögren’s reticular dystrophy can be diagnosed by ophthalmoscopy when the pigment accumulated in the retinal RPE resembles a fishing net with distinct knots. These lesions are usually bilateral and symmetrical and should be confirmed by autofluorescence, fluorescein angiography and electrophysiological tests (mfERG, ERG, EOG) with normal or discreetly blurred visual acuity and the absence of other visual disorders associated with retinal function demonstrated in electrophysiological studies. A typical characteristic is the extension of the lesions from the center to the periphery of the retina with a progressive reduction of the grid pattern. These changes are accompanied by normal or slightly subnormal electroretinographic results. The authors identified all such features in the studied patient. 

It should be noted that the EOG test result is not always significantly disturbed in young patients, as in the described case. There exists the possibility of carriage in the pathogenic variant in the region of the C2 gene-c.841_849+19del (dbSNP rs9332736), which may be important in the development of this disease, which has not been mentioned in the literature so far. Patients should be closely and regularly monitored for the potential progression of dystrophy, and choroidal neovascularization may be one symptom. It is interesting that may be, in this type of genetic base, multifocal ERG may reveal slight dysfunction macular cones photoreceptor and bipolar bioelectrical abnormality. Significant asymmetry between eyes and lowering Arden ratio in the left eye (3.3/1.8) was shown with electrooculography (EOG) and may suggest early bioelectrical dysfunction of RPE/photoreceptors layers in this case. Flash ERG (ERG) revealed only slight increases in implicit times of the a and b waves of the rod-cone response. Obtained results of ERG exludes with high probability cone and cone-rod dystrophy. 

Subtle ERG and EOG changes in a patient with Sjogren’s reticular dystrophy with the described pathogenetic variant require a careful ophthalmological approach and systematic follow-up to assess the prognosis for future vision.

## Figures and Tables

**Figure 1 jcm-12-01406-f001:**
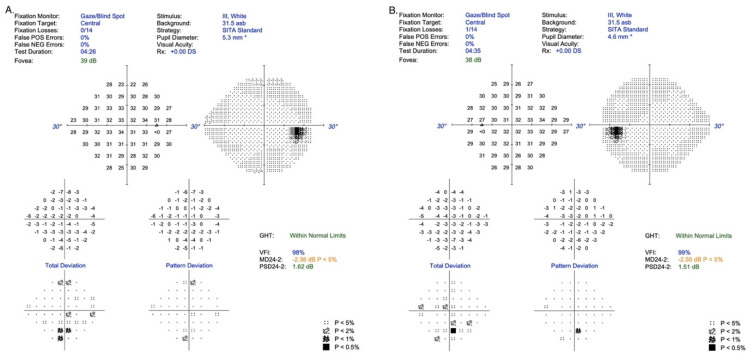
Static perimetry, (**A**) Right eye; (**B**) Left eye.

**Figure 2 jcm-12-01406-f002:**
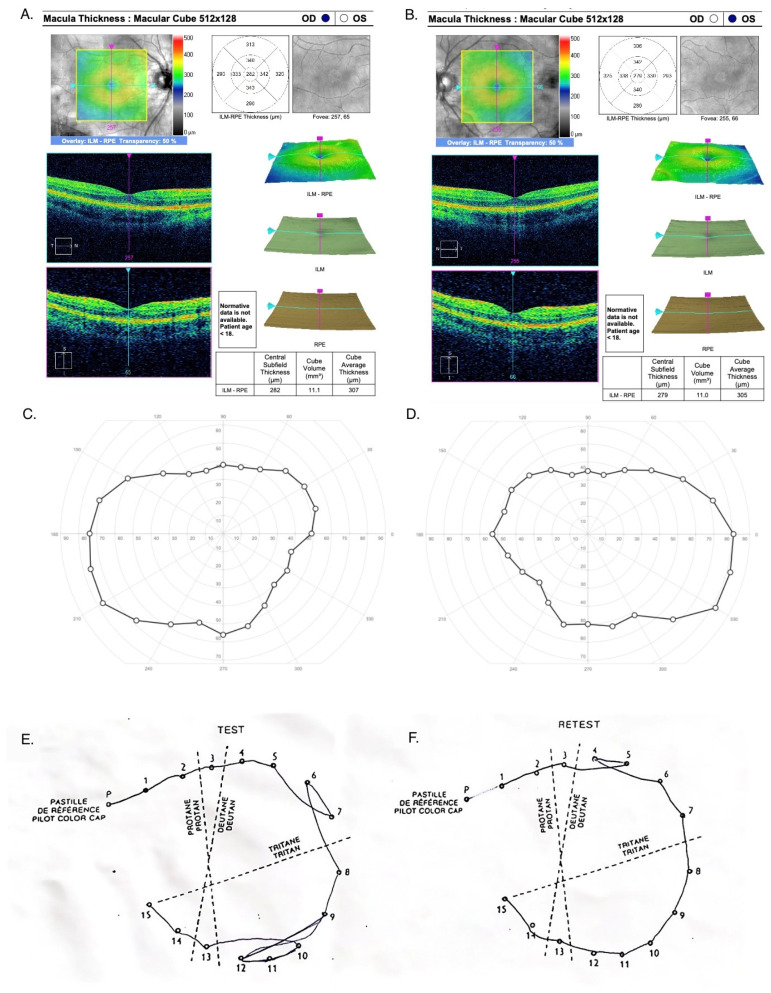
Correct recording in the OCT, RE (**A**) and LE (**B**); kinetic perimetry, RE (**C**) and LE (**D**); and Farnsworth D-15 tests, RE (**E**) and LE (**F**).

**Figure 3 jcm-12-01406-f003:**
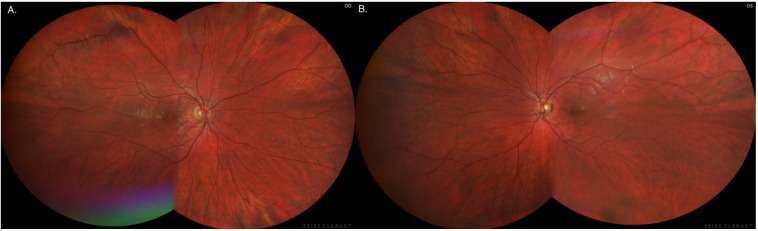
Fundus imaging in the right eye (**A**) and left eye (**B**), respectively.

**Figure 4 jcm-12-01406-f004:**
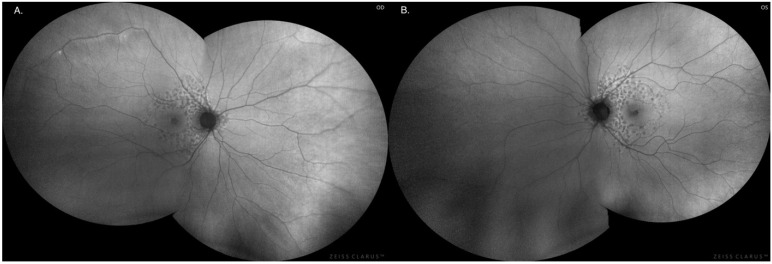
Autofluorescence of the fundus of the right eye (**A**) and left eye (**B**), respectively, with characteristic RPE lesions located centrally and in the middle periphery of the retina, visualized with the characteristic reticular pattern.

**Figure 5 jcm-12-01406-f005:**
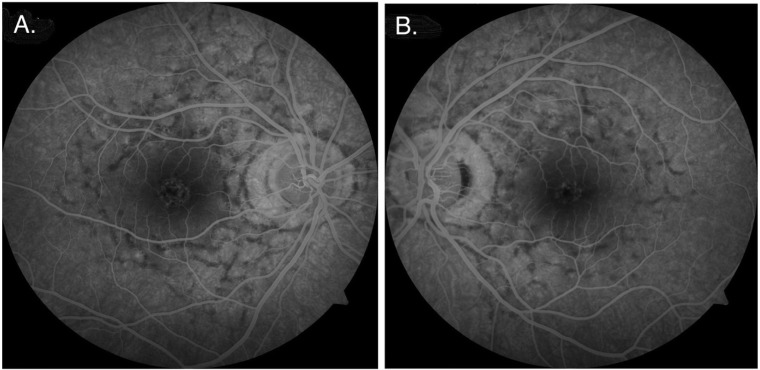
Fluorescein angiography imaging, right eye (**A**) and left eye (**B**), respectively.

**Figure 6 jcm-12-01406-f006:**
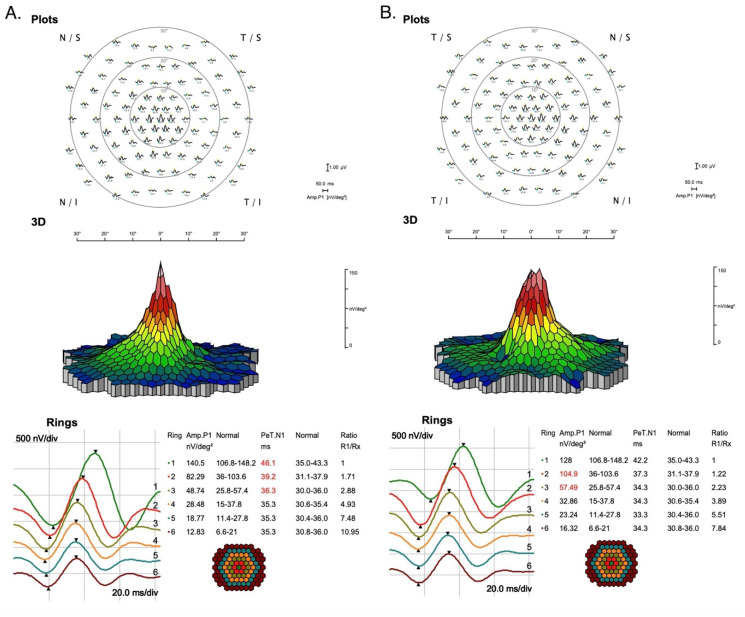
MfERG and impaired functional activity of cones/bipolars in right (**A**) and left eyes (**B**).

**Figure 7 jcm-12-01406-f007:**
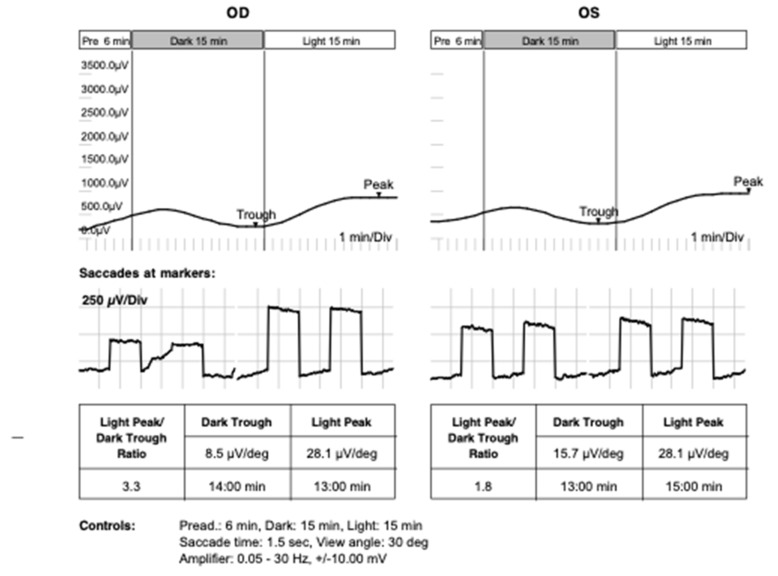
EOG test result in a patient with reticular dystrophy. Near lower limit of normal Arden ratio in the left eye was detected.

**Figure 8 jcm-12-01406-f008:**
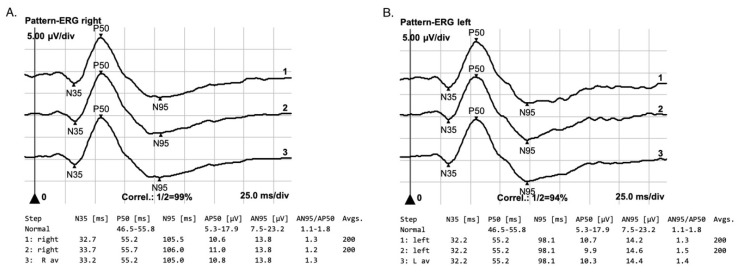
Normal PERG-test results in right eye (**A**) and left eye (**B**).

## Data Availability

The data presented in this study is available on request from the corresponding author. The data is not publicly available due to privacy reasons.

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
