# Peer review of "Observations for Sjögren’s Pigment Epithelial Reticular Dystrophy in a 16-Year-Old Boy—An Extremely Rare Retinal Case Report"

_jcm, 2023, doi:10.3390/jcm12041406_

Round 1
Reviewer 1 Report (Previous Reviewer 2)
Dear Authors,
I wish to submit my review of the article: "Observations toward Sjögren's pigment epithelial reticular dystrophy in a 16-year old boy – an extremely rare retinal case report."
The authors well addressed my previous queries.
Author Response
The authors of the article would like to thank again the reviewer for their helpful comments which raised the rank of the text and made the article interesting for the recipient-reader.
Best regards,
Monika Modrzejewska
Reviewer 2 Report (New Reviewer)
The current manuscript is an extremely rare retinal case report aiming to present observations toward the Sjögren's pigment epithelial reticular dystrophy in a 16-year-old boy. Although the topic is significant in the field of ocular clinical medicine, there are several issues that definitely require the authors’ attention to improve the quality of this particular manuscript.
Specific comments:
1. As stated by the authors, Sjögren’s reticular dystrophy is an extremely rare disease. Therefore, this case report is unique. Nevertheless, from the viewpoint of ocular clinical medicine, the pathological mechanism of Sjögren’s reticular dystrophy is unknown and raises the readers’ curiosity. The authors should describe pertinent information in detail.
2. As stated by the authors, hyperpigmentation, after spreading towards the periphery of the retina, tended to disappear in later years. If the pathological conditions will spontaneously be resolved with time, the audiences are unaware of the importance and necessity of diagnosis of Sjögren's pigment epithelial reticular dystrophy. Please further specify.
3. According to my checking, the authors indeed describe the patient’s history and diagnosis. But, the audiences are curious about whether the subject have received any treatment to relieve the visual disturbance?
4. In fact, many treatment modalities have been developed for the treatment of retinal diseases. More recently, investigators have developed new ocular nanotherapeutics with multiple bioactive properties for potential use in the management of retinal disorders (DOI: 10.1021/acsnano.2c05824), which should be illustrated to balance the scientific viewpoint and attract more attention from audiences. Although the current report aims to explore the diagnostics of retinal funcitons for ocular clinical medicine, the authors are highly recommended to consider the inclusion of this relevant paper in the reference list to broaden and deepen the Discussion section of the article content.
Author Response
- In response to the reviewer, I would like to inform you that the missing information has been supplemented and the text has been extended with the content requested by the reviewer.
2. Information about the movement of the dye in the RPE, which disappears over time - in the later years of the patient from places - where it was present to new places in the retina, has been supplemented quite extensively. This was supported by relevant quotations. this is indeed the characteristic retinal pigmentation seen in these dystrophies, as confirmed by the authors we quote.
3. As we mentioned, the patient described by us had no visual disturbances, so he did not require treatment - however, he requires ophthalmological observation, which is important because we confirmed in the patient a strictly defined pathogenic variant that may be associated with changes in the neurosensory retina giving changes in the ERG - which is rarely described in this types of RPE dystrophies. We have also included this information in the abstract and in the conclusions.
- According to the reviewer's suggestions - for which I thank you kindly, I included the position on nanotherapeutics to the information on the possibilities of modern medicine as a variant of possible therapies in dystrophies that damage vision
We would also like to mention that we submitted the work to an English grammar check to improve the overall image of the work.
The authors of the article would like to thank the reviewers for their helpful comments which raised the rank of the text and made the article interesting for the recipient-reader.
Best regards,
Monika Modrzejewska

Round 2
Reviewer 2 Report (New Reviewer)
The revised version has adequately addressed most of the critiques raised by this reviewer and is now suitable for publication in "JCM".
This manuscript is a resubmission of an earlier submission. The following is a list of the peer review reports and author responses from that submission.
Round 1
Reviewer 1 Report
In this paper, the authors presented a rare case with the Sjögren's pigment epithelial reticular dystrophy. They described characteristic of fundus morphology, fluorescein angiography, mfERG electrophysiology, and primarily electrooculogram tests. But there are still several issues need to be addressed.
1. The Figures did not include autofluorescence and OCT exam result. It will be better to systematically describe the characteristic of a rare case.
2. Did the case receive any treatment? And what's the follow up or outcome?
3. Sjögren’s reticular dystrophy can be inherited in an autosomal dominant or recessive pattern. Did the case have any family history? Did the authors consider to take a genetic test to confirm it?
4. Figure 1,3 4 were from scanned hard copies. Please provide high quality pictures.
Author Response
For the 1st reviewer
Thank you for the positive assessment and the questions asked.
- As suggested, the remaining accounts that had not been deleted have withdrawn. Together with those tests that are valid. I systematized the order of the presented photos.
The results of the oct autofluorescence studies were included and listed in the correct order.2. The patient does not require treatment at the moment because his visual acuity is correct despite such significant changes visible in additional ophthalmic examinations3. Sjögren's reticular dystrophy may be inherited in an autosomal dominant or recessive manner. Due to the high cost of genetic tests, we did not perform them. However, we are considering such a possibility -as Hereditary retinal dystrophy and retinal dystrophy syndromes. Screening of approximately 300 genes using next-generation NGS sequencing methods.
- Figures 1,3 4 are actually poor resolution - for which we apologize, and I am sending photos in much better resolution, which should be better received by the evaluator. Sorry again
Thank you for the questions asked, which always introduce valuable comments and enrich the text. Regards Monika Modrzejewska

Reviewer 2 Report
Dear Authors,
I wish to submit my review for the paper titled: "Sjögren's pigment epithelial reticular dystrophy in a 16-year-old boy – an extremely rare retinal case report."
Sjögren's pigment epithelial reticular dystrophy is a rare disease, and the Authors should be commended for their work and deep analysis.
However, some points need to be proofread:
1. English Language requires moderate proofreading. For example, many abbreviations are not reported in the text.
2. Introduction: Could you please amend and expand it to provide a 360 degrees background according to current Literature?
3. Case presentation: Did you perform any molecular analysis? Could you Please amend the abbreviations?
4. All figures lack the caption. (the caption is in the supplementary material, it should be placed before the pictures). Could you please add OCT, infrared, and autofluorescence images?
5. Discussion: Despite the extensive literature discussion, Could you please deeply focus on your case, describing your findings according to the current Literature?
(If available): Do you have information regarding the follow-up and actions that were taken? It could be interesting to analyze the disease evolution further. (If available, please add further details regarding it).
Author Response
For the 2-nd reviewer
Thank you for the positive assessment and the questions asked.
- Improved the English language to the extent of the native speaker's capabilities. Abbreviations of words or names used in the text have been inserted.
- Lack of current literature in the world - only single cases of this disease are described. So it is difficult to broaden the introduction. But if the reviewer has such suggestions, we will try to deepen the topic of retinal dystrophy in general. Please respond in this regard.
3. Sjögren's reticular dystrophy may be inherited in an autosomal dominant or recessive manner. Due to the high cost of genetic tests, we did not perform them. However, we are considering such a possibility -as Hereditary retinal dystrophy and retinal dystrophy syndromes. Screening of approximately 300 genes using next-generation NGS sequencing methods.
4. Of course thanks for your comments, all photos and tests done on the patient are included, including vinegar and autofluorescence and the quality of these photos is improved.
5. In the text - we have included references analyzing the results of the obtained research in relation to the results of other authors which are available in a very small number of items in the journal. Unfortunately, there is no extensive knowledge in this case.
Our observation is the onset of the disease without damaging the visual function. The changes are only clinical. Perhaps in the future, functional changes will be captured.
Thank you for the questions asked, which always introduce valuable comments and enrich the text. Regards Monika Modrzejewska

Round 2
Reviewer 1 Report
In their revised manuscript, the authors have responded to the comments provided in the original one. But there are still several issues need to be addressed.
It will be better to use electronic images instead of scanned hard copies.
Family history should be added in case presentation.
Author Response
Author's Reply to the Review Report (Reviewer 1)
Thank you for your favorable opinion on our article - an interesting and rare case-report. We've done our best to improve the quality of a few photos that were copied - and we've tried to improve the quality of the photos again. Currently, digital versions have been used - so these are the highest quality images that we were able to present. The only figure that needs to remain in hard scanned copy form is the Fanworth D-15 test as we do not have it in digital form. In response to the inquiry regarding ophthalmic changes in the patient's family, I can only say that there are no current results of ophthalmic examinations. Recent results did not show any lesions at the fundus - however, not all tests necessary to diagnose sjogren's syndrome have been performed. If I were to include a family history now, I have to do all the necessary tests and invite these people as patients - which will certainly not be easy considering their work, activities, not necessarily their interest in their disease, the time they have to spend, and the travel costs they have to near. But of course maybe I can do it. But I may also try to do this in the future and present the fundus appearance of the same family members in Sjogren's syndrome, which can be very varied and even slightly marked ... which is characteristic of this disease. I also understand that this is a limitation of this article. Thank you for the effort you put into reviewing the work. Regards Monika ModrzejewskaReviewer 2 Report
Dear Authors, I wish to submit my review of the manuscript you have previously revised.
I carefully evaluated the revised version of the manuscript, finding it substantially improved.
If available, Could you please add any information regarding the familiar medical history?
Author Response
Author's Reply to the Review Report (Reviewer 2)
Thank you for looking positively on my effort and your good appreciation. In line with earlier comments, I corrected the version of the description by supplementing it with detailed results of additional research and improving the resolution of the photos sent. Another suggestion for a family medical history - has been extended and supplemented as far as possible - because we use the medical data that is recorded in the patient's record. This complements our description of the patient. In response to the inquiry regarding ophthalmic changes in the patient's family, I can only say that there are no current results of ophthalmic examinations. Recent results did not show any lesions at the fundus - however, not all tests necessary to diagnose sjogren's syndrome have been performed. If I were to include a family history now, I have to do all the necessary tests and invite these people as patients - which will certainly not be easy considering their work, activities, not necessarily their interest in their disease, the time they have to spend, and the travel costs they have to near. But of course maybe I can do it. But I may also try to do this in the future and present the fundus appearance of the same family members in Sjogren's syndrome, which can be very varied and even slightly marked ... which is characteristic of this disease.I also understand that this is a limitation of this article. Thank you for the effort you put into reviewing the work. Regards Monika Modrzejewska